# Detection of 13 Ginsenosides (Rb1, Rb2, Rc, Rd, Re, Rf, Rg1, Rg3, Rh2, F1, Compound K, 20(*S*)-Protopanaxadiol, and 20(*S*)-Protopanaxatriol) in Human Plasma and Application of the Analytical Method to Human Pharmacokinetic Studies Following Two Week-Repeated Administration of Red Ginseng Extract

**DOI:** 10.3390/molecules24142618

**Published:** 2019-07-18

**Authors:** Sojeong Jin, Ji-Hyeon Jeon, Sowon Lee, Woo Youl Kang, Sook Jin Seong, Young-Ran Yoon, Min-Koo Choi, Im-Sook Song

**Affiliations:** 1College of Pharmacy, Dankook University, Cheon-an 31116, Korea; 2College of Pharmacy and Research Institute of Pharmaceutical Sciences, Kyungpook National University, Daegu 41566, Korea; 3Clinical Trial Center, Kyungpook National University Hospital, Daegu 41944, Korea; 4Department of Biomedical Science, BK21 Plus KNU Bio-Medical Convergence Program for Creative Talent, College of Medicine, Kyungpook National University, Daegu 41944, Korea

**Keywords:** ginsenosides, red ginseng extract, pharmacokinetics, human

## Abstract

We aimed to develop a sensitive method for detecting 13 ginsenosides using liquid chromatography–tandem mass spectrometry and to apply this method to pharmacokinetic studies in human following repeated oral administration of red ginseng extract. The chromatograms of Rb1, Rb2, Rc, Rd, Re, Rf, Rg1, Rg3, Rh2, F1, compound K (CK), protopanaxadiol (PPD), and protopanaxatriol (PPT) in human plasma were well separated. The calibration curve range for 13 ginsenosides was 0.5–200 ng/mL and the lower limit of quantitation was 0.5 ng/mL for all ginsenosides. The inter- and intra-day accuracy, precision, and stability were less than 15%. Among the 13 ginsenosides tested, nine ginsenosides (Rb1, Rb2, Rc, Rd, Rg3, CK, Rh2, PPD, and PPT) were detected in the human plasma samples. The plasma concentrations of Rb1, Rb2, Rc, Rd, and Rg3 were correlated with the content in red ginseng extract; however, CK, Rh2, PPD, and PPT were detected although they are not present in red ginseng extract, suggesting the formation of these ginsenosides through the human metabolism. In conclusion, our analytical method could be effectively used to evaluate pharmacokinetic properties of ginsenosides, which would be useful for establishing the pharmacokinetic–pharmacodymic relationship of ginsenosides as well as ginsenoside metabolism in humans.

## 1. Introduction

Ginsenosides are classified into two types according to their hydroxylation position on the core triterpene saponin structure: 20(*S*)-protopanaxadiol (PPD) and 20(*S*)-protopanaxatriol (PPT) [1]. These ginsenosides are considered to be the major active pharmacological constituents of ginseng [2,3]. Several studies have described the immunological, antioxidant, anticoagulant, anti-neoplastic, neuroprotective, and hepatoprotective effects of ginseng and its associated ginsenosides [3,4,5,6,7,8]. The content and types of ginsenosides vary depending on the preparation method of ginseng product such as steaming times, temperature, and the extraction method [9,10]. For example, ginsenosides Rg1 and Re decreased, but ginsenosides Rb1, Rb2, Rc, Rd, and Rg3 increased after several hours of steaming and extraction. As results, the ratio of PPD-type to PPT-type ginsenoside of Korean red ginseng extract was higher than that of Korean ginseng [10]. Ginsenosides generally have low intestinal permeability in Caco-2 cells and low oral bioavailability in rats [1]. For example, the oral bioavailability of Rb1 and Rh2 is around 1.18–4.35 % and 4.0–6.4%, respectively. Other ginsenosides such as Rg1, Rd, Rh1, and Re have a low oral bioavailability, of less than 10% [1]. Owing to the low oral bioavailability of these ginsenosides, their plasma concentration is also low. The maximum plasma concentration of Rb1, Rb2, Rc, and Rd, major ginsenosides found in rat plasma, was lower than 10 ng/mL in rats following oral administration of red ginseng extract at a dose of 1.5 g/kg [8,11]

Because of the low plasma concentration of ginsenosides, the analysis of ginsenosides in human plasma following oral administration of ginseng product has been limited to the selected ginsenoside. Moreover, analytical methods have also been limited to liquid chromatography–tandem mass spectrometry (LC-MS/MS) rather than high-performance liquid chromatography (HPLC) with UV or fluorescence detection [12]. In human study, there are some reports on the analysis of ginsenosides but they used large volume of plasma or provided limited concentrations on ginsenosides because of the high lower limit of quantitation (LLOQ). For example, the plasma concentration of Rb1 and CK following single oral administration of 10 g of American ginseng powder was investigated. In this study, Rb1 and CK in 0.7 mL of plasma samples were extracted using a solid-phase extraction procedure and detected by time-of-flight mass spectrometry coupled with ultra-high pressure liquid chromatography [13]. Another study developed a simultaneous analysis method for Rb1 and Rg1 in human plasma by LC-MS/MS. In this study, for the analysis of Rb1 and Rg1, 100 μL of human plasma was subjected to protein precipitation and analyzed with a calibration curve range of 10–1000 ng/mL [14]. The ginsenoside Rb1 was detected but Rg1 was not detected. This could be attributed to the low plasma concentration of Rg1 after oral administration of the ginseng product (1.5 g/day) [14].

Recently, more sensitive analytical methods have been developed. Choi et al. reported the plasma concentrations of Rb1 and CK in human plasma following single oral administration of 3 g of fermented red ginseng extract with a calibration curve range of 1–1000 ng/mL [15]. In another study, ginsenoside PPD was analyzed after a single oral administration of a PPD 25 mg capsule with a calibration curve range of 0.1–100 ng/mL [16]. Ginsenoside Re was analyzed after a single oral administration of a Re 200 mg tablet with a calibration curve range of 0.5–200 ng/mL. The metabolite peaks of Rg1, Rg2, F1, Rh1, and PPT in human plasma and urine were also monitored following oral administration of Re tablet (200 mg) without quantification [12]. Our group simultaneously determined the plasma concentrations of the ginsenosides Rb1, Rb2, Rc, Rd, and CK in human subjects following single and 2-week repeated administration of three pouches of red ginseng product with a calibration curve range of 0.5–200 ng/mL [17].

However, minor ginsenosides or metabolites of ginsenosides may also have beneficial pharmacological effects and, therefore, the pharmacokinetic properties of minor components and metabolites should also be measured. In rats, following single or repeated oral administration of high doses of ginseng extract (2–8 g/kg), various ginsenosides such as Ra3, Rb1, Rd, CK, Re, and Rg1 could be detected in the plasma of rats by LC-MS/MS with calibration curves ranging from 1.37 or 12.3 ng/mL to 3000 ng/mL [18]. These results suggest that sensitive analytical methods could be useful for the detection of various ginsenosides in human plasma.

Therefore, the objective of this study was to develop an analytical method for the detection of various ginsenosides in human plasma and to apply this validated method to pharmacokinetic studies after multiple administration of red ginseng extract (three pouches/day for two weeks) in human subjects. We analyzed 13 ginsenosides (Rb1, Rb2, Rc, Rd, Re, Rf, Rg1, Rg3, Rh2, F1, CK, PPD, and PPT), which are ginsenosdies found in red ginseng extract and their biological metabolites that could be transformed by intestinal microbiota (Figure 1).

## 2. Results

### 2.1. MS/MS Analysis

The mass spectrometer was operated with electrospray ionization (ESI) in the positive ionization mode. Table 1 shows the selected precursor and product ions of analytes and respective mass spectrometric conditions in the MS/MS stage of the ginsenosides, which were optimized based on the fragmentation patterns of precursor and product ions of target ginsenoside, the specificity of target ginsenoside compared to the other ginsenosides, and the consistency with the previously published findings [11,19]. Since ginsenosides Rb2 and Rc resulted in the same *m/z* values of precursor and product ion, these ginsenosides should be separated each other during the elution. Retention times were 5.7 min for Rb2 and 4.8 min for Rc (Table 1 and Figure 2B).

### 2.2. Sample Praperations

For sample preparation, both protein precipitation and liquid–liquid extraction (LLE) methods should be applied depend on the number of glycosylation of ginsenosides. For example, we used the protein precipitation method for ginsenosides glycosylated with more than two glucose units (i.e., Rb1, Rb2, Rc, Rd, Re, Rf, Rg1, Rg3, and F2; hydrophilic ginsenosides) and the LLE method for monoglycosylated ginsenosides and their aglycones (i.e., Rh1, Rh2, CK, PPD, and PPT; lipophilic ginsenosides) based on the extraction recovery after sample preparation and the interference of endogenous peaks in human blank plasma (the plasma withdrawn from human subjects who did not take ginseng or ginsenosides). The monoglycosylated ginsenoside F1 could be extracted with both the protein precipitation and LLE method; however, the detection sensitivity of analyte was better for precipitation samples than for LLE samples. Therefore, F1 were extracted with the protein precipitation method. Methyl tert-butyl ether (MTBE) was chosen as an extraction solvent based on the extraction efficiency and reproducibility of the ginsenosides Rh1, Rh2, CK, PPD, and PPT and based on previous findings [15].

The ginsenosides F2 and Rh1 were excluded in the validation process because their peaks could not be completely separated from the endogenous peaks that detected at the same *m/z* as F2 and Rh1 in human blank plasma, and the peak response of F2 and Rh1 at LLOQ was less than five times the response of a blank sample [20,21].

### 2.3. Analytical Method Validation

The method was fully validated according to the FDA Guidance for Industry: Bioanalytical Method Validation (May 2018) [21] for its specificity, accuracy, precision, matrix effect and extraction recovery, and stability.

#### 2.3.1. Specificity

Representative multiple reaction-monitoring (MRM) chromatograms of the ginsenosides Rb1, Rb2, Rc, Rd, Re, Rf, Rg1, Rg3, Rh2, F1, F2, CK, PPD, and PPT (Figure 2) showed that all the ginsenoside peaks obtained using the protein precipitation or LLE method were well separated with no interfering peaks at their respective retention times. The retention times of the 13 ginsenosides are shown in Table 1. The specificity of the analytes was confirmed using six different human blank plasma samples and test plasma samples obtained from human subjects at 1 h after the last oral administration of red ginseng extract (Figure 2).

#### 2.3.2. Linearity and LLOQ

To assess linearity, the standard calibration curve of eight different concentrations of 13 ginsenosides was analyzed, and the standard calibration curve and equation for each component are shown in Table 2. The LLOQ was defined as a signal-to-noise ratio of > 5.0 with a precision rate of ≤ 15% and an accuracy rate of 80–120%. The LLOQ for the ginsenosides in our analytical system was set at 0.5 ng/mL in all cases.

#### 2.3.3. Precision and Accuracy

The inter-day and intra-day precision and accuracy were assessed using three different concentrations (1.5, 15, and 150 ng/mL) of quality control (QC) samples consisting of a specific ginsenoside mixture (Rb1, Rb2, Rc, Rd, Re, Rf, Rg1, Rg3, and F1 for protein precipitation; Rh2, CK, PPD, and PPT for LLE) (Table 3). The results showed that inter-day and intra-day precision (CV in Table 3) for the 13 ginsenosides was below 13.0%, and the inter-day and intra-day accuracy (RE in Table 3) for the 13 ginsenosides was below 15.0% (Table 3).

#### 2.3.4. Extraction Recovery and Matrix Effect

The extraction recovery of the ginsenosides Rb1, Rb2, Rc, Rd, Re, Rf, Rg1, Rg3, and F1, which were prepared with the protein precipitation method using three different concentrations (1.5, 15, and 150 ng/mL) of QC samples, ranged from 85.5% to 99.2% with a CV of < 14.9%. In the case of the LLE method, the extraction recovery of the ginsenosides Rh2, CK, PPD, and PPT ranged from 56.3% to 81.9% with a CV of < 14.9% (Table 4).

The matrix effects for the ginsenosides Rh2, CK, PPD, and PPT ranged from 77.0% to 100.1%. The matrix effects for the protein-precipitated ginsenosides (Rb1, Rb2, Rc, Rd, Re, Rf, Rg1, Rg3, and F1) ranged from 7.0% to 92.9%. The matrix effect of ginsenosides Re, Rf, and Rg1 was in the range of 7.0%–19.5%, suggesting that Re, Rf, and Rg1 showed significant signal suppression during the ionization and protein precipitation process; however, the values of CV of Re, Rf, and Rg1 was less than 15% and the matrix effect of Re, Rf, and Rg1 was similar for the three different QC levels with an acceptable CV, and 10 other ginsenosides showed no significant interference during ionization and sample preparation. According to the EMA guideline [22], we concluded our analytical method was acceptable even though Re, Rf, and Rg1 had significant ion suppression.

#### 2.3.5. Stability

The precision (CV) and accuracy (RE) of three different concentrations of QC samples consisting of a mixture of the ginsenosides Rb1, Rb2, Rc, Rd, Re, Rf, Rg1, Rg3, and F1, which were prepared using the protein precipitation method, were within 13.5% for short-term stability, below 14.9% for post-preparative stability, and below 12.9% for three freeze–thaw cycle stability (Table 5). The precision (CV) and accuracy (RE) of three different concentrations of QC samples consisting of a mixture of the ginsenosides Rh2, CK, PPD, and PPT, which were prepared using the LLE method, were within 10.6% for short-term stability, below 12.4% for post-preparative stability, and below 14.7% for three freeze–thaw cycle stability (Table 5). Therefore, the 13 ginsenosides in human plasma samples had no stability issues during the storage in the freezer, sample preparation process, and analysis time after the samples were processed, as demonstrated by the three stability tests.

### 2.4. Contents of Ginsenosides in Red Ginseng Extract

The ginsenoside content of the red ginseng extract provided to participants daily for 14 days (three pouches of Hongsamjung All Day^TM^/day) is summarized in Table 6. The most abundant ginsenoside was Rb1 (18.8–23.6 mg/day), followed by Rb2, Rc, Rd, and Rg3 (12.9–5.9 mg/day). The abundance of Re, Rh1, and Rg1 was 1.6–6.6 mg/day. The daily intake of PPT-type ginsenosides was lower than that of PPD-type ginsenosides. The values of daily intake of PPD-type ginsenosides are ranged between 50.2–64.7 mg/day and those of PPT-type ginsenoside are ranged between 11.2–14.9 mg/day.

The oral administration of three pouches of red ginseng for two weeks was well tolerated and did not produce any unexpected or serious adverse events, as previously reported [17].

### 2.5. Pharmacokinetics of Rb1, Rb2, Rc, Rd, Rg3, Rh2, CK, PPD, and PPT Following 2 Weeks-Repeated Administration of Red Ginseng Extract

Of the 13 ginsenosides examined, nine ginsenosides (Rb1, Rb2, Rc, Rd, Rg3, CK, Rh2, PPD, and PPT) were detected in the human plasma samples; the plasma concentrations of these ginsenosides are shown in Figure 3. The ginsenosides Rb1, Rb2, Rc, Rd, and Rg3, which were detected in the plasma samples, are all PPD-type ginsenosides and present at a relatively high content in red ginseng extract. In contrast, the PPT-type ginsenosides Re and Rh1 were not detected in the human plasma samples despite their high content in red ginseng extract. CK, Rh2, and PPD, which are metabolites from Rb1, Rb2, and Rc, were also detected even though they are not present in red ginseng extract, suggesting that these PPD-type metabolites could be formed in the human intestine during the intestinal absorption stage (Figure 1) [11,23,24]. Among the reported PPT-type metabolites, only PPT was detected in the human plasma.

The pharmacokinetic parameters from the plasma concentration-time profiles of these ginsenosides are shown in Table 7. The plasma Rb1, Rb2, Rc, and Rd concentrations were constant over time, and they had a long terminal half-life. The AUC and C_max_ values of Rb1, Rb2, Rc, and Rd were correlated with the content of red ginseng extract. In contrast to the plasma concentrations of Rb1, Rb2, Rc, and Rd, the plasma concentrations of Rg3, Rh2, and CK showed a bell-shaped profile (Figure 3); this may be attributed to further metabolism to PPD. The T_max_ of Rg3 (3.6 h) was smaller than that of Rh2 and CK (5.6–9.1 h), which may be associated with the high content of Rg3 that the absorption of Rg3 could occur following oral administration of red ginseng extract and absence of Rh2 and CK in the red ginseng extract that the absorption of Rh1 and CK could occur after they were transformed from Rb1, Rb2, Rc, and Rd.

The plasma concentration profiles of PPD and PPT were similar but flatter compared with those of Rg3, Rh2, and CK. Since PPD was derived from Rg3, Rh2, and CK and could undergo further metabolism [11,23,24], the plasma profile of PPD and PPT could be attributed to the faster elimination in human body rather than intestinal formation via intestinal microbiota. Lin et al. reported that 40 metabolites of PPD were identified in human plasma and urine and the major metabolites of PPD was the hydroxylated form in human body through phase I hepatic metabolism [19].

To explain time-dependent metabolism and absorption of ginsenosides, the plasma concentrations of ginsenosides at absorption phase (from 4 to 10 h) depend on the deglycosylation states was shown in Figure 4. The plasma concentrations of Rb1, Rb2, Rc, and Rd, tri- and tetraglycosylated ginsenosides, were stable for 4–10 h of post dose (Figure 4A), suggesting the stable absorption and slow elimination process. The plasma concentrations of Rg3 was decreased along with increasing time (4–10 h) but the monoglycosylated ginsenosides Rh2 and CK, metabolites from Rg3 and F2, increased over time (Figure 4B,C), suggesting the gut metabolism from Rg3 to Rh2 during the absorption stage. The delayed absorption of Rh2, CK, and PPD indicated that formation and absorption of Rh2, CK, and PPD might occur in the lower part of intestine. On the other hand, the formation and absorption of PPT was faster than PPD (Figure 4D), suggesting the rapid metabolism of PPT-type ginsenosides in human intestine and it partly attributed to the absence of Re and Rg1 in human plasma despite of the higher content in Korean red ginseng extract.

## 3. Discussion

Despite the therapeutic benefits of various ginsenosides, which include anti-cancer, anti-diabetic, anti-oxidative, and immune-stimulating effects [3,4,5,6,7,8], the plasma concentration of these ginsenosides and their pharmacokinetic-pharmacodynamic relationship need to be further investigated. As its first step, analytical methods for various ginsenosides and pharmacokinetic profile of these ginsenosides are critical. We developed an analytical method for 13 ginsenosides (Rb1, Rb2, Rc, Rd, Re, Rf, Rg1, Rg3, and F1, Rh2, CK, PPD, and PPT) using a LC-MS/MS system, which had high sensitivity (i.e., the LLOQ of all ginsenosides was 0.5 ng/mL) and required a small plasma sample volume (100 μL). The glycosylation number of the ginsenosides was different: tetraglycosylated ginsenosides for Rb1, Rb2, and Rc; triglycosylated ginsenosides for Rd, Re, and Rg1; diglycosylated ginsenosides for F2, Rg3, and Rf; monoglycosylated ginsenosides for Rh2, CK, Rh1, and F1; aglycones for PPD and PPT (Figure 1). Because of different extraction efficiencies, di-, tri-, and tetraglycosylated ginsenosides were extracted by protein precipitation, and aglycones were extracted by LLE. Monoglycosylated ginsenosides could be extracted using both methods; however, CK and Rh2 were extracted by LLE, and F1 was extracted by protein precipitation based on the extraction recovery and matrix effect.

We further validated our sensitive analytical method by performing a pharmacokinetic study after the oral administration of red ginseng extract (three pouches of red ginseng extract), which has demonstrated tolerability for two weeks of repeated administration [17]. We successfully measured the plasma concentration of Rb1, Rb2, Rc, Rd, Rg3, Rh2, CK, PPD, and PPT. Except for PPT, detectable ginsenosides were all PPD-type ginsenosides and their deglycosylated metabolites. Interestingly, the plasma AUC values of three glycosylated ginsenosides (Rb1, Rb2, and Rc) were correlated with the content of red ginseng extract and showed similar T_max_ values, suggesting the similar intestinal absorption kinetics of these ginsenosides despite of the different structures and glycosidation patterns, which is consistent with the previous report [17]. The long terminal half-life suggested that the intestinal metabolism (to other PPD-type metabolites) and excretion of Rb1, Rb2, and Rc may be a slow process. The T_max_ values of Rd, Rh2, CK, and PPD were increased according to the deglycosylated status, suggesting that deglycosylation mediated by β-glucosidase in the intestinal microbiome could occur sequentially and steadily [11,23,24], and Rh2, CK, and PPD could be detected in human plasma even though they are not present in red ginseng extract.

In the case of Rg3, its T_max_ was smaller compared with that of Rh2 and CK because of its high content in red ginseng extract. Re and Rg1 (PPT-type ginsenosides) were not detected even though they are present in red ginseng extract; however, PPT was detected. It is possible that Re and Rg1 are metabolized to PPT by intestinal microbiota before the absorption occur [11,23,24] and biotransformation of PPT could be faster than the formation rate of PPD. However, we should note that the time-dependent gut metabolism of ginsenosides in human intestine has never been investigated, therefore we speculated time-dependent gut metabolism of ginsenoside from the plasma concentration and T_max_ of ginsenosides and their deglycosylated metabolites. Particularly, for CK concentration, large inter-subject variation was shown in Figure 4B and previous publication [17]. This variability could be attributed to inter-subject variable metabolism related to the intestinal microbiota [25] and further studies should focus on the characterization of microorganisms that produce it and the potential beneficial effects of this metabolite.

## 4. Materials and Methods

### 4.1. Materials

Red ginseng extract (Hongsamjung All Day^TM^; lot no. 731902) was purchased from the Punggi Ginseng Cooperative Association (Youngjoo, Kyungpook, Republic of Korea). The ginsenosides Rb1, Rb2, Rc, Rd, Re, Rf, Rg1, Rg3, Rh1, Rh2, F1, F2, CK, PPD, and PPT were purchased from the Ambo Institute (Daejeon, Republic of Korea). Berberine and 13C-caffeine, used as internal standards (IS), were purchased from Sigma-Aldrich Chemical Co. (St. Louis, MO, USA). All other chemicals and solvents were of reagent or analytical grade.

### 4.2. LC-MS/MS Analysis

#### 4.2.1. Instrument

The LC-MS/MS system consisted of an Agilent 1260 Infinity HPLC system (Agilent Technologies, Wilmington, DE, USA) and Agilent 6470 Triple Quadrupole MS system (Agilent Technologies, Wilmington, DE, USA). The system was operated using Mass Hunter Acquisition Software (Version B.08.00; Agilent Technologies, Wilmington, DE, USA). The pressure of drying gas was set at 35 psi and the gas temperature was kept at 300 °C. The ion spray voltage was set at 4000 V in the positive mode.

#### 4.2.2. HPLC Condition

Chromatographic separation was performed using a Phenomenex Polar RP analytical column (150 × 2.0 mm i.d., 4.0 μm particle size) for protein precipitation samples and a Phenomenex Luna C18 analytical column (150 × 2.0 mm i.d., 3.0 μm particle size) for liquid–liquid extraction (LLE) samples. The HPLC mobile phase for protein precipitation samples consisted of 0.1% formic acid in water (phase A) and 0.1% formic acid in methanol (phase B), and the following gradient elution was used: 69% of phase B for 0–2.0 min, 69–85% of phase B for 2.0–4.0 min, 85–69% of phase B for 6.0–6.5 min. The flow rate was 0.27 mL/min, and the injection sample volume was 10 μL. The HPLC mobile phase for LLE samples was isocratic, consisting of 0.1% formic acid in water (8%) and 0.1% formic acid in methanol (92%) at a flow rate of 0.15 mL/min. The sample injection volume was 10 μL.

#### 4.2.3. Preparation of Stock, Working, and Quality Control (QC) Solutions

Ginsenosides and their metabolites (Rb1, Rb2, Rc, Rd, Re, Rf, Rg1, Rg3, Rh2, F1, CK, PPD, and PPT) were accurately weighed and dissolved in methanol to obtain a concentration of 1000 μg/mL each.

The above stock solutions were divided and mixed according to the sample preparation method (i.e., protein precipitation and LLE). The ginsenosides for protein precipitation method (Rb1, Rb2, Rc, Rd, Re, Rf, Rg1, Rg3, and F1) were mixed and diluted with methanol to a concentration of 2000 ng/mL. The ginsenosides for LLE (Rh2, CK, PPD, and PPT) were mixed and diluted with methanol to a concentration of 2000 ng/mL. Working solutions were then serially diluted with methanol to obtain calibration working solutions of 5, 10, 20, 50, 200, 500, 1000, and 2000 ng/mL. Quality control (QC) working solutions were prepared at 15, 150, and 1500 ng/mL with each ginsenoside.

#### 4.2.4. Preparation of Calibration Curve and QC Samples

Calibration curve samples were prepared by spiking 10 μL of working solution into 90 μL of human blank plasma at final concentrations of 0.5, 1, 2, 5, 20, 50, 100, and 200 ng/mL. QC samples were prepared by spiking 10 μL of QC working solution into 90 μL of human blank plasma at final concentrations of 1.5, 15, and 150 ng/mL of QC samples.

For protein precipitation, 600 μL of an IS (0.05 ng/mL berberine in methanol) was added to 100 μL of calibration curve samples and QC samples. Then, the mixture was vortexed for 15 min and centrifuged at 16,100× *g* for 5 min. After centrifugation, 500 μL of the supernatant was transferred to a clean tube and evaporated to dryness under a nitrogen stream at 40 °C. The residue was reconstituted with 150 μL of 70% methanol consisting of 0.1% formic acid.

For LLE, 50 μL of an IS (20 ng/mL 13C-caffeine in water) and 800 μL of MTBE was added to 100 μL of calibration curve samples and QC samples. The mixture was vortexed for 10 min and centrifuged at 16,100× *g* for 5 min. After centrifugation, the samples were frozen at −80 °C for 4 h. The upper layer was transferred to a clean tube and evaporated to dryness under a nitrogen stream. The residue was reconfigured with 150 μL of 80% methanol consisting of 0.1% formic acid.

### 4.3. Method Validation

#### 4.3.1. Specificity

The specificity of the method was assessed by comparing chromatogram responses of six lots of human blank plasma with lower limit of quantification (LLOQ) sample.

#### 4.3.2. Linearity

The linearity of the method was assessed using six calibration curves analyzed on six different days. The calibration curve was obtained by plotting the peak area ratio against the concentration of each drug at eight-point levels with a weighting factor of 1/x^2^.

#### 4.3.3. Precision and Accuracy

The intra-day (*n* = 5) and inter-day (*n* = 6) precision and accuracy were evaluated using three different QC samples for each analyte. The precision and accuracy at each concentration level were evaluated in terms of the coefficient of variance (CV, %) and relative error (RE, %).

#### 4.3.4. Extraction Recovery and Matrix Effect

The extraction recovery and matrix effect were assessed for three different QC samples using six different blank plasma samples. The extraction recoveries were evaluated by comparing the peak areas of the extracted samples (spiked before extraction) with those of the unextracted samples (spiked after blank extraction) [26]. The matrix factor for the analyte and IS was calculated in each lot by comparing the peak responses of the post-extraction samples (spiked after blank extraction) against neat solutions, which have the same amount of analyte as the extracted sample [26].

#### 4.3.5. Stability

Short-term stability was evaluated to determine whether the sample was stable during treatment. All analytes and IS of the spiked plasma samples were left for at least 6 h at 25 °C. The spiked plasma samples were also subjected to a freeze (−80 °C) and thaw cycle (25 °C and stand for 2 h) three times. After the samples were processed, it was confirmed that they were stable at 8 °C for 24 h. The stability test was conducted using three different concentrations of QC samples.

### 4.4. Pharmacokinetic Study

The study was approved by the Institutional Review Board of Kyungpook National University Hospital (KNUH, Daegu, Republic of Korea) and was conducted at the KNUH Clinical Trial Center in accordance with the applicable Good Clinical Practice guidelines (IRB approval no. KNUH 2018-04-028-002). All subjects provided written informed consent before study enrollment and underwent clinical evaluation including physical examination, serology tests, 12-lead electrocardiography, and clinical history assessment. A total of 11 healthy Korean male subjects aged ≥ 19 years and with a body weight of ≥ 50 kg were enrolled in this study.

The volunteers took 3 pouches of red ginseng extract per day at 9 AM for 2 weeks. On the 14th day, after taking the last dose of the red ginseng extract, blood samples (5 mL) were collected in a heparinized tube at 0.25, 0.5, 1, 2, 3, 4, 6, 8, 10, 12, and 24 h post-dose via a saline-locked angiocatheter. The plasma was collected by centrifugation for 10 min at 3000 × g and stored at −80 °C until analysis.

To analyze the ginsenosides Rb1, Rb2, Rc, Rd, Re, Rf, Rg1, Rg3, and F1, 600 μL of an IS (0.05 ng/mL berberine in methanol) was added to 100 μL of plasma samples. Then, the mixture was vortexed for 15 min and centrifuged at 16,100× *g* for 5 min. After centrifugation, 500 μL of the supernatant was transferred to a clean tube and evaporated to dryness under a nitrogen stream at 40 °C. The residue was reconstituted with 150 μL of 70% methanol consisting of 0.1% formic acid, and a 10 μL aliquot was injected into the LC-MS/MS system.

To analyze the ginsenosides Rh2, CK, PPD, and PPT, 50 μL of an IS (20 ng/mL 13C-caffeine in water) and 800 μL of MTBE were added to 100 μL of plasma samples. The mixture was vortexed for 10 min and centrifuged at 16,100× *g* for 5 min. After centrifugation, the samples were frozen at −80 °C for 4 h. The upper layer was transferred to a clean tube and evaporated to dryness under a nitrogen stream. The residue was reconfigured with 150 μL of 80% methanol consisting of 0.1% formic acid, and a 10 μL aliquot was injected into the LC-MS/MS system.

Similarly, the ginsenoside content in the red ginseng extract was quantified. The red ginseng extract (100 mg) was diluted 50-fold with methanol, and 100 μL of the diluted sample was prepared using the method described previously. Aliquots (10 μL) of the supernatant were directly injected into the LC-MS/MS system.

### 4.5. Data Analysis

Pharmacokinetic parameters were estimated using non-compartmental methods (WinNonlin version 2.0; Pharsight Co., Certara, NJ, USA). All pharmacokinetic parameters are presented as the mean ± standard deviation (SD).

## 5. Conclusions

A sensitive LC–MS/MS method for the detection of 13 ginsenosides (Rb1, Rb2, Rc, Rd, Re, Rf, Rg1, Rg3, and F1, Rh2, CK, PPD, and PPT) in human plasma with a LLOQ of 0.5 ng/mL was developed and validated. This method can be used in the bioanalysis and pharmacokinetic studies of ginseng products administered at multiple therapeutic doses. Following repeated oral administration of red ginseng extract for two weeks, the plasma concentrations of Rb1, Rb2, Rc, Rd, Rg3, Rh2, CK, PPD, and PPT were detected. The findings can provide valuable information on ginsenoside metabolism in the human body and contribute to in vivo pharmacokinetic-pharmacodynamic correlation studies.

## Figures and Tables

**Figure 1 molecules-24-02618-f001:**
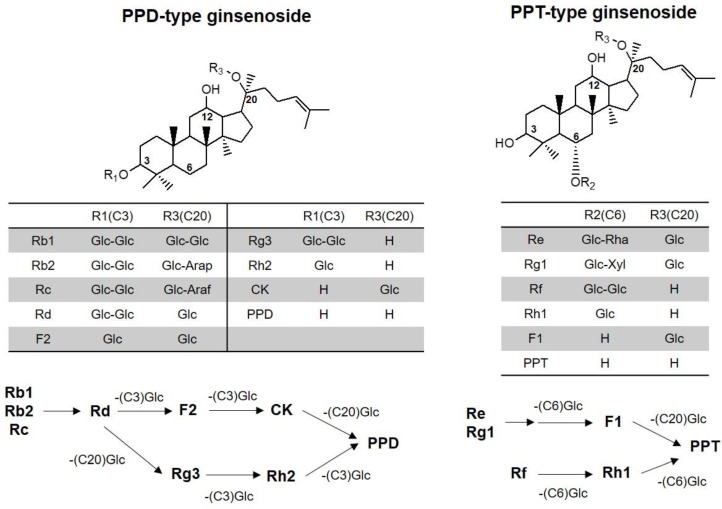
Structure and metabolic pathway of 20(*S*)-protopanaxadiol (PPD) and 20(*S*)-protopanaxatriol (PPT) type ginsenosides. Metabolic pathway represents deglycosylation at C3, C6, or C20 position by β-glucosidase from intestinal microbiota. Glc: glucose; Arap: arabinopyranose; Araf: arabinofuranose; Rha: rhamnose; Xyl: xylose.

**Figure 2 molecules-24-02618-f002:**
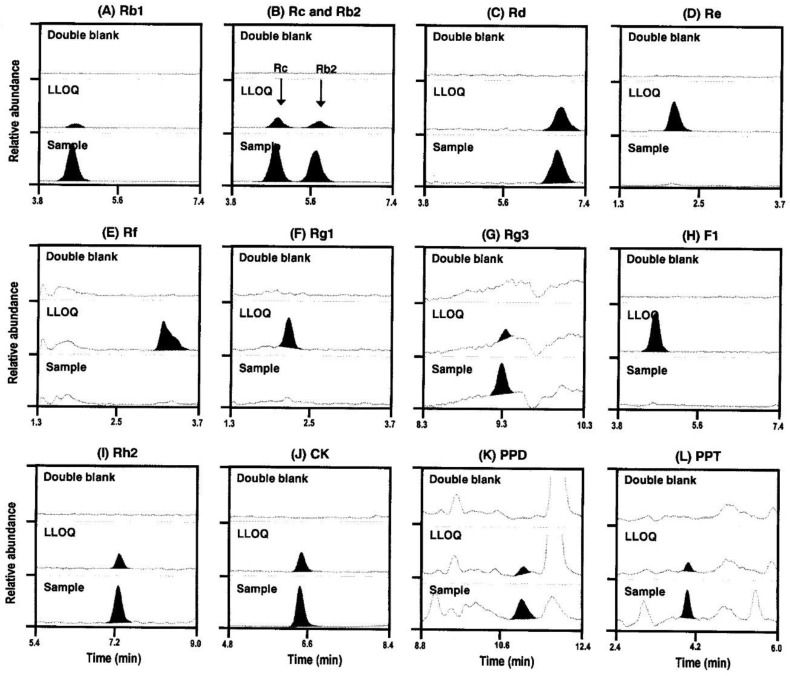
Representative multiple reaction-monitoring (MRM) chromatograms of the ginsenosides (**A**) Rb1, (**B**) Rc and Rb2, **(C)** Rd, (**D**) Re, (**E**) Rf, (**F**) Rg1, (**G**) Rg3, (**H**) F1, (**I**) Rh2, (**J**) CK, (**K**) PPD, and (**L**) PPT in human double blank plasma (**upper**), human blank plasma spiked with standard samples with a lower limit of quantification (LLOQ) (**center**), and human plasma at 1 h following 2 weeks of repeated oral administration of red ginseng extract (**lower**).

**Figure 3 molecules-24-02618-f003:**
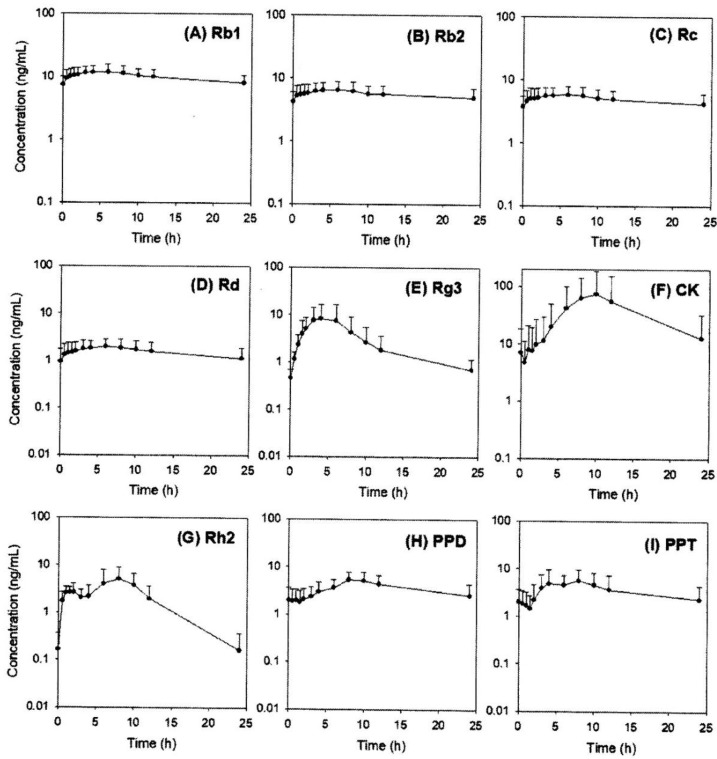
Plasma concentration-time profiles of ginsenosides (**A**) Rb1, (**B**) Rb2, (**C**) Rc, (D) Rd, (**E**) Rg3, (**F**) CK, (**G**) Rh2, (**H**) PPD, and (**I**) PPT in human plasma after two-weeks repeated administrations of red ginseng extract. Data represented as mean ± SD from eleven subjects.

**Figure 4 molecules-24-02618-f004:**
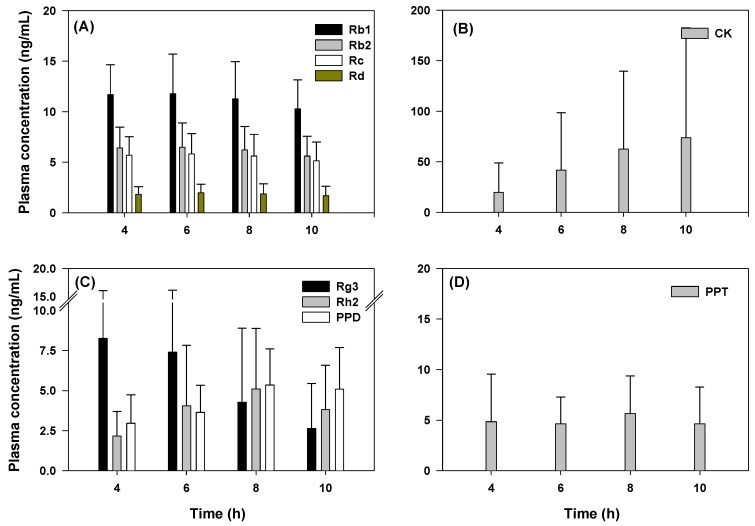
Plasma concentration of ginsenosides (**A**) Rb1, Rb2, Rc, and Rd, (**B**) CK, (**C**) Rg3, Rh2, and PPD, and (**D**) PPT at 4, 6, 8, and 10 h after two-weeks repeated administrations of red ginseng extract. Data represented as mean ± SD from eleven subjects.

**Table 1 molecules-24-02618-t001:** Mass spectrometry (MS/MS) parameters for the detection of the ginsenosides and internal standard (IS).

Sample Preparation Method	Compound	Precursor Ion (*m/z*)	Product Ion (*m/z*)	Retention Time (min)	Fragmentor Voltage (V) ^a^	Collision Energy (V)
Protein precipitation	Rb1	1131.6	365.1	4.6	165	65
	Rb2	1101.6	335.1	5.7	185	60
	Rc	1101.6	335.1	4.8	185	60
	Rd	969.9	789.5	6.8	170	50
	Re	969.9	789.5	2.1	170	50
	Rf	823.5	365.1	3.3	135	55
	Rg1	824.0	643.6	2.2	135	40
	Rg3	807.5	365.2	9.3	165	60
	F1	661.5	203.1	4.6	185	40
	F2	807.5	627.5	9.4	135	40
	Berberine (IS)	336.1	320.0	4.5	135	30
LLE	Rh1	603.4	423.4	2.9	135	10
	Rh2	587.4	407.4	4.5	135	15
	CK	645.5	203.1	6.4	160	35
	PPD	425.3	109.1	11.0	125	25
	PPT	441.3	109.1	4.0	130	30
	13C-caffeine (IS)	198	140	2.9	120	20

^a^ Fragmentor voltage (V) is the voltage difference between capillary and skimmer.

**Table 2 molecules-24-02618-t002:** Linear range, slope and intercept of regression equation, and correlation coefficient of 13 ginsenosides.

Analyte	Linear Range (ng/mL)	Slope ± SD ^a^	Intercept ± SD ^a^	Correlation Coefficient ^a^
Rb1	0.5–200	0.0485 ± 0.0205	0.0007 ± 0.0019	0.997
Rb2	0.5–200	0.1069 ± 0.0394	−0.0003 ± 0.0041	0.997
Rc	0.5–200	0.1408 ± 0.0393	0.0003 ± 0.0047	0.997
Rd	0.5–200	0.2597 ± 0.0536	−0.0388 ± 0.0903	0.996
Re	0.5–200	0.2509 ± 0.0290	−0.0048 ± 0.0095	0.997
Rf	0.5–200	0.1980 ± 0.0308	−0.0056 ± 0.0095	0.995
Rg1	0.5–200	0.0648 ± 0.0081	−0.0010 ± 0.0071	0.994
Rg3	0.5–200	0.0687 ± 0.0092	0.0008 ± 0.0069	0.987
F1	0.5–200	0.8728 ± 0.2221	−0.0337 ± 0.0437	0.995
Rh2	0.5–200	0.0146 ± 0.0035	−0.0006 ± 0.0009	0.996
CK	0.5–200	0.0860 ± 0.0442	−0.0071 ± 0.0076	0.990
PPD	0.5–200	0.0476 ± 0.0120	0.0004 ± 0.0024	0.995
PPT	0.5–200	0.0221 ± 0.0022	−0.0019 ± 0.0037	0.996

^a^ Average of six determinations.

**Table 3 molecules-24-02618-t003:** Intra- and inter-day precision and accuracy of 13 ginsenosides.

Analyte	QC (ng/mL)	Inter-day (*n* = 5)				Intra-day (*n* = 6)			
		Measured (ng/mL)	SD	CV (%)	RE (%)	Measured (ng/mL)	SD	CV (%)	RE (%)
Rb1	1.5	1.5	0.1	6.1	0.9	1.5	0.1	7.7	1.9
	15	15.1	0.7	4.5	0.5	15.0	0.3	1.9	0.2
	150	150.3	8.8	5.9	0.2	154.8	5.8	3.7	3.2
Rb2	1.5	1.5	0.1	4.7	2.5	1.5	0.1	5.2	1.4
	15	15.0	0.6	4.2	−0.1	15.2	0.3	2.2	1.4
	150	154.5	10.3	6.6	3.0	162.6	6.6	4.1	8.4
Rc	1.5	1.5	0.1	4.4	−0.8	1.5	0.1	5.0	0.3
	15	14.9	0.8	5.0	−0.4	14.8	0.5	3.2	−1.3
	150	154.1	11.9	7.7	2.7	160.2	7.6	4.7	6.8
Rd	1.5	1.5	0.1	6.4	2.0	1.6	0.1	5.8	6.3
	15	14.9	0.7	5.0	−0.8	15.2	0.3	2.1	1.2
	150	155.8	10.7	6.9	3.9	164.4	6.4	3.9	9.6
Re	1.5	1.5	0.1	3.7	−0.7	1.6	0.1	3.9	6.1
	15	15.1	0.6	4.2	0.5	15.4	0.5	3.1	2.8
	150	150.1	9.8	6.5	0.1	158.6	6.0	3.8	5.7
Rf	1.5	1.5	0.1	6.0	0.4	1.6	0.1	5.2	4.4
	15	15.0	0.6	3.7	−0.3	15.3	0.9	5.6	2.2
	150	156.7	11.4	7.3	4.5	166.3	7.9	4.8	10.8
Rg1	1.5	1.6	0.1	6.1	3.9	1.6	0.1	5.9	6.2
	15	15.2	0.9	5.8	1.5	15.8	0.5	2.9	5.1
	150	151.1	7.7	5.1	0.7	155.5	5.6	3.6	3.7
Rg3	1.5	1.4	0.2	10.3	−5.9	1.6	0.1	5.9	8.5
	15	15.8	1.6	10.2	5.2	15.9	1.2	7.5	5.9
	150	159.2	16.0	10.1	6.1	166.4	11.7	7.0	11.0
F1	1.5	1.5	0.1	6.5	−3.3	1.5	0.1	3.7	−0.3
	15	14.9	0.8	5.5	−1.0	14.9	0.3	2.2	−0.7
	150	154.8	10.9	7.1	3.2	160.8	7.4	4.6	7.2
Rh2	1.5	1.4	0.1	8.2	−4.9	1.3	0.1	9.5	−11.8
	15	15.2	0.4	2.6	1.5	13.6	1.2	8.6	−9.2
	150	151.5	7.5	5.0	1.0	151.2	16.9	11.2	0.8
CK	1.5	1.4	0.2	10.6	−4.9	1.5	0.1	6.5	−3.4
	15	14.5	1.9	13.0	−3.5	12.8	1.4	10.8	−15.0
	150	163.2	12.0	7.3	8.8	141.1	10.0	7.1	−6.0
PPD	1.5	1.5	0.2	11.7	−1.1	1.5	0.1	5.1	3.0
	15	14.9	0.5	3.5	−0.5	15.0	0.5	3.2	−0.1
	150	166.4	17.2	10.3	10.9	155.4	6.3	4.1	3.6
PPT	1.5	1.5	0.1	5.0	2.4	1.5	0.1	4.2	1.4
	15	14.8	0.4	2.8	−1.5	15.7	0.4	2.4	4.6
	150	153.8	2.0	1.3	2.5	156.8	2.3	1.5	4.5

Data represented as mean ± SD from five or six independent experiments.

**Table 4 molecules-24-02618-t004:** Extraction recoveries and matrix effects for 13 ginsenosides.

Analyte	QC (ng/mL)	Recovery (%)		Matrix Effect (%)		Analyte	QC (ng/mL)	Recovery (%)		Matrix Effect (%)	
		Recovery	CV	Matrix Effect	CV			Recovery	CV	Matrix Effect	CV
Rb1	1.5	93.5	8.5	74.5	13.6	Rg3	1.5	91.9	14.9	92.2	12.2
	15	85.5	8.1	78.3	7.0		15	90.3	6.4	70.3	6.8
	150	89.3	4.0	75.8	5.8		150	94.2	10.2	67.9	4.2
Rb2	1.5	96.0	6.5	79.4	8.5	F1	1.5	96.9	7.2	56.6	4.0
	15	86.0	5.9	82.1	6.1		15	90.1	6.5	57.9	3.2
	150	89.3	2.9	78.5	6.3		150	93.1	3.3	57.1	3.2
Rc	1.5	93.8	6.7	69.8	12.5	Rh2	1.5	64.9	11.8	99.7	3.2
	15	87.7	6.4	70.4	8.9		15	64.9	3.7	95.6	2.4
	150	91.3	4.3	67.8	8.4		150	65.4	3.6	98.5	2.6
Rd	1.5	96.4	7.3	73.9	8.0	CK	1.5	60.0	14.9	88.6	7.3
	15	88.7	5.2	75.5	8.4		15	64.0	14.4	93.1	6.6
	150	90.8	3.8	72.4	6.8		150	56.3	12.4	77.0	5.1
Re	1.5	99.2	13.1	9.5	7.4	PPD	1.5	79.9	5.0	98.4	14.9
	15	93.5	5.8	7.0	5.5		15	70.7	5.4	96.7	4.1
	150	95.5	3.1	7.9	6.6		150	71.6	5.2	100.1	5.7
Rf	1.5	93.0	4.2	19.5	13.6	PPT	1.5	81.7	7.2	77.8	11.8
	15	88.1	6.7	16.4	9.9		15	77.5	4.3	77.3	5.3
	150	94.0	7.1	18.2	10.9		150	81.9	6.8	76.6	5.4
Rg1	1.5	97.5	13.4	9.9	5.8						
	15	97.6	8.5	7.2	3.4						
	150	96.2	5.5	7.7	4.2						

Data represented as mean ± SD from six independent experiments.

**Table 5 molecules-24-02618-t005:** Stability of 13 ginsenosides.

**Analyte**	**Short-Term Stability (6 h, 25 °C)**				**Analyte**	**Short-Term Stability (6 h, 25 °C)**			
	**QC (ng/mL)**	**Measured (ng/mL)**	**CV (%)**	**RE (%)**		**QC (ng/mL)**	**Measured (ng/mL)**	**CV (%)**	**RE (%)**
Rb1	1.5	1.6	5.3	3.9	Rg3	1.5	1.4	13.5	-7.9
	15	13.5	3.3	−9.9		15	13.7	4.7	−8.7
	150	137.0	6.8	−8.7		150	133.7	4.2	−10.8
Rb2	1.5	1.5	1.6	−2.9	F1	1.5	1.5	2.3	−2.9
	15	13.3	4.7	−11.5		15	13.3	4.5	−11.1
	150	137.5	8.7	−8.3		150	138.9	6.4	−7.4
Rc	1.5	1.4	2.8	−4.3	Rh2	1.5	1.5	5.1	0.2
	15	13.4	3.4	−10.6		15	14.9	2.9	−0.8
	150	138.5	8.2	−7.6		150	149.7	4.3	−0.2
Rd	1.5	1.5	4.0	−3.5	CK	1.5	1.4	4.7	−6.5
	15	13.2	4.2	−12.2		15	13.4	1.5	−10.6
	150	139.7	8.0	−6.9		150	143.7	7.9	−4.2
Re	1.5	1.5	1.8	−1.3	PPD	1.5	1.6	5.0	5.4
	15	14.1	2.0	−5.8		15	15.3	2.2	2.0
	150	143.4	3.7	−4.4		150	146.2	5.6	−2.5
Rf	1.5	1.4	3.0	−7.6	PPT	1.5	1.5	3.0	1.0
	15	13.6	3.4	−9.1		15	15.0	2.0	−0.3
	150	150.3	9.2	0.2		150	146.0	5.0	−2.7
Rg1	1.5	1.5	5.6	2.0					
	15	14.0	3.1	−6.9					
	150	144.9	4.7	−3.4					
**Analyte**	**Post-Preparative Stability (24 h, 8 °C)**				**Analyte**	**Post-Preparative Stability (24 h, 8 °C)**			
	**QC (ng/mL)**	**Measured (ng/mL)**	**CV (%)**	**RE (%)**		**QC (ng/mL)**	**Measured (ng/mL)**	**CV (%)**	**RE (%)**
Rb1	1.5	1.5	14.9	−0.3	Rg3	1.5	1.4	9.7	−8.7
	15	13.9	5.3	−7.5		15	14.3	5.3	−4.5
	150	133.3	2.9	−11.1		150	131.2	3.1	−12.5
Rb2	1.5	1.4	10.1	−9.3	F1	1.5	1.3	4.9	−12.4
	15	13.6	7.7	−9.5		15	13.5	6.7	−10.1
	150	135.3	3.4	−9.8		150	134.9	1.8	−10.1
Rc	1.5	1.4	9.7	−5.5	Rh2	1.5	1.5	5.2	2.0
	15	13.6	6.1	−9.7		15	15.6	2.5	3.7
	150	135.4	2.8	−9.8		150	164.8	3.1	9.9
Rd	1.5	1.3	5.4	−11.3	CK	1.5	1.4	12.4	−5.9
	15	13.4	7.0	−10.7		15	13.7	4.9	−8.8
	150	136.2	3.0	−9.2		150	160.7	3.3	7.2
Re	1.5	1.5	5.1	−2.6	PPD	1.5	1.6	2.0	5.0
	15	14.2	4.8	−5.4		15	15.5	3.0	3.1
	150	141.7	0.6	−5.5		150	156.9	1.8	4.6
Rf	1.5	1.7	1.9	14.6	PPT	1.5	1.7	4.5	10.8
	15	14.8	2.7	−1.2		15	15.7	2.7	4.5
	150	154.2	7.3	2.8		150	158.2	2.7	5.5
Rg1	1.5	1.4	2.5	−4.0					
	15	14.2	9.7	−5.3					
	150	142.8	3.3	−4.8					
**Analyte**	**Freeze-Thaw Stability (3 Cycles)**				**Analyte**	**Freeze-Thaw Stability (3 Cycles)**			
	**QC (ng/mL)**	**Measured (ng/mL)**	**CV (%)**	**RE (%)**		**QC (ng/mL)**	**Measured (ng/mL)**	**CV (%)**	**RE (%)**
Rb1	1.5	1.5	2.3	−2.6	Rg3	1.5	1.6	2.7	9.2
	15	13.6	3.2	−9.5		15	13.6	1.4	−9.2
	150	143.6	5.4	−4.3		150	140.1	4.7	−6.6
Rb2	1.5	1.4	4.3	−4.7	F1	1.5	1.5	3.1	−0.7
	15	13.3	2.2	−11.1		15	13.4	1.4	−10.4
	150	145.2	5.4	−3.2		150	145.8	5.0	−2.8
Rc	1.5	1.5	1.7	−0.3	Rh2	1.5	1.5	4.0	0.4
	15	13.5	2.1	−10.3		15	15.6	4.7	3.7
	150	147.8	5.8	−1.5		150	152.3	2.1	1.5
Rd	1.5	1.4	2.0	−5.7	CK	1.5	1.5	7.8	−3.3
	15	13.1	2.3	−12.9		15	13.1	14.7	−12.4
	150	144.7	5.4	−3.6		150	167.3	4.7	11.5
Re	1.5	1.5	2.4	0.4	PPD	1.5	1.6	6.0	6.6
	15	13.9	2.4	−7.1		15	15.7	3.7	4.5
	150	144.5	5.8	−3.7		150	160.4	2.2	7.0
Rf	1.5	1.4	1.3	−5.5	PPT	1.5	1.4	5.6	−3.5
	15	14.0	2.9	−6.8		15	15.4	1.6	2.8
	150	156.8	7.7	4.5		150	158.7	1.7	5.8
Rg1	1.5	1.5	3.1	−2.5					
	15	14.1	4.6	−6.3					
	150	147.3	3.1	−1.8					

Data represented as mean ± SD from six independent experiments.

**Table 6 molecules-24-02618-t006:** Daily intake amount of ginsenoside from red ginseng extract.

Ginsenoside		mg/day	Ginsenoside		mg/day
PPD-type	Rb1	21.9 ± 2.1	PPT-type	Re	6.6 ± 1.3
	Rb2	10.4 ± 1.2		Rg1	5.2 ± 0.6
	Rc	12.9 ± 1.5		F1	0.0 ± 0.0
	Rd	5.9 ± 0.7		PPT	0.0 ± 0.0
	Rh2	0.0 ± 0.0			
	Rg3	7.9 ± 2.3			
	CK	0.0 ± 0.0			
	PPD	0.0 ± 0.0			

Data represented as mean ± SD from four independent experiments.

**Table 7 molecules-24-02618-t007:** Pharmacokinetic parameters of ginsenosides in human plasma after two-weeks repeated administrations of red ginseng extract.

Ginsenosides	PK Parameters				
	AUC (ng∙h/mL)	C_max_ (ng/mL)	T_max_ (h)	MRT (h)	T_1/2_ (h)
Rb1	227.6 ± 73.5	12.7 ± 3.6	4.5 ± 1.8	10.7 ± 1.5	42.9 ± 20.8
Rb2	137.0 ± 48.8	6.9 ± 2.3	4.5 ± 2.3	11.8 ± 1.5	51.2 ± 22.8
Rc	123.0 ± 46.1	6.2 ± 2.1	4.3 ± 3.2	11.7 ± 1.5	34.5 ± 12.9
Rd	35.1 ± 19.5	2.2 ± 0.9	6.2 ± 2.1	10.4 ± 1.5	24.6 ± 8.0
Re	ND	ND	ND	ND	ND
Rf	ND	ND	ND	ND	ND
Rg1	ND	ND	ND	ND	ND
Rg3	68.0 ± 60.5	8.7 ± 8.9	3.6 ± 0.9	8.2 ± 1.4	9.4 ± 3.9
F1	ND	ND	ND	ND	ND
Rh2	49.9 ± 27.8	6.1 ± 3.5	6.0 ± 3.3	7.7 ± 1.5	3.1 ± 1.3
CK	873.0 ± 1236.0	81.6 ± 112.5	9.5 ± 1.6	10.6 ± 1.2	5.2 ± 1.1
PPD	85.1 ± 39.5	6.1 ± 2.3	8.7 ± 1.6	11.3 ± 1.9	12.6 ± 8.2
PPT	86.5 ± 49.8	7.9 ± 4.6	8.3 ± 6.2	11.2 ± 3.0	10.6 ± 8.4

AUC: area under the plasma concentration-time curve from 0 to last sampling time. C_max_: maximum plasma concentration; T_max_: time to reach C_max_; MRT: mean residence time. T_1/2_: half-life; ND: not detected. Data represented as mean ± SD from eleven subjects

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
