# Peer review of "Detection of 13 Ginsenosides (Rb1, Rb2, Rc, Rd, Re, Rf, Rg1, Rg3, Rh2, F1, Compound K, 20(S)-Protopanaxadiol, and 20(S)-Protopanaxatriol) in Human Plasma and Application of the Analytical Method to Human Pharmacokinetic Studies Following Two Week-Repeated Administration of Red Ginseng Extract"

_molecules, 2019, doi:10.3390/molecules24142618_

Reviewer 1 Report

This is a very interesting work that can help many other researchers working in the in vivo validation of potentially bioactive compounds. 

The manuscript is very well written and figures and tables are very helpful except for Figure 3. Why the initial concentrations do not start closer to 0? Additionally, the most relevant information of this figure is included in table 7. Instead it would be very intersting to show comparative chromatograms at 4, 6, 8 and 10 hours (values close to Tmax of the different compounds). Particularly, for CK concentration it would be important to discuss about the variability observed since it is a metabolite related to microbiota, further studies should focus on the characterization of microorganisms that produce it and the potential beneficial effects of this metabolite. 

Finally, please change the word microflora for microbiota. 

Author Response

Q1. The manuscript is very well written and figures and tables are very helpful except for Figure 3. Why the initial concentrations do not start closer to 0? Additionally, the most relevant information of this figure is included in table 7. Instead it would be very interesting to show comparative chromatograms at 4, 6, 8 and 10 hours (values close to Tmax of the different compounds). Particularly, for CK concentration it would be important to discuss about the variability observed since it is a metabolite related to microbiota, further studies should focus on the characterization of microorganisms that produce it and the potential beneficial effects of this metabolite.

Answer> We analyzed ginsenoside in human plasma following 2-weeks of repeated administration of red ginseng. Therefore, the concentrations of ginsenoside at zero time represent the trough level of ginsenoside before the administration of last ginseng dose and it was not close to 0 because of long half-life of ginsenoside.

As the reviewer suggested, we added a Figure 4 to explain time-dependent metabolism and absorption of ginsenosides, the plasma concentrations of ginsenosides at absorption phase (from 4 to 10 h) depend on the deglycosylation states was shown in Figure 4. The plasma concentrations of Rb1, Rb2, Rc, and Rd, tri- and tetraglycosylated ginsenosides, were stable for 4 - 10 h of post dose (Figure 4A), suggesting the stable absorption and slow elimination process. The plasma concentrations of Rg3 was decreased along with increasing time (4 – 10 h) but the monoglycosylated ginsenosides Rh2 and CK, metabolites from Rg3 and F2, increased over time (Figure 4B and 4C), suggesting the gut metabolism from Rg3 to Rh2 during the absorption stage. The delayed absorption of Rh2, CK, and PPD indicated that formation and absorption of Rh2, CK, and PPD might occur in the lower part of intestine. On the other hand, the formation and absorption of PPT was faster than PPD (Figure 4D), suggesting the rapid metabolism of PPT-type ginsenosides in human intestine and it partly attributed to the absence of Re and Rg1 in human plasma despite of the higher content in Korean red ginseng extract. However, we should note that the time-dependent gut metabolism of ginsenosides in human intestine has never been investigated, therefore we speculated time-dependent gut metabolism of ginsenoside from the plasma concentration and Tmax of ginsenosides and their deglycosylated metabolites. Particularly, for CK concentration, large inter-subject variation was shown in Figure 4B and previous publication. This variability could be attributed to inter-subject variable metabolism related to the intestinal microbiota and further studies should focus on the characterization of microorganisms that produce it and the potential beneficial effects of this metabolite.

We did not delete Figure 3 because the pharmacokinetic profiles of ginsenosides were thought to be important. Instead, Figure 4, related results, and discussion were added during the revision [Lines 241-252 and 300-308]. We asked the reviewer’s generous understanding on this issue.     

Q2. Please change the word microflora for microbiota.

Answer> According to the reviewer’s comment, we have changed microflora to microbiota in page line.

Reviewer 2 Report

The paper is well-organized and well-written. The research design is appropriate and well-realized. The methods are adequately described and the results are clearly presented.

The paper could be accepted in the present form.

Author Response

Thank you for your positive response.

Reviewer 3 Report

The paper is about quantification of ginesosides in human plasma by an analytical method and pharmacokinetic studies

The paper is interesting showing the evolution of the compounds in human metabolism. However, there are some critical points that need to be clarified or corrected before publication:

The methodology is robust but, the discussion and conclusion sections are weak. Aims and goals of the work are clearly described.

Abstrac: have 255 words more than 200 as the instructions of author indicates.

Line 29 I suggest write the LOQ limits as a range of the highest to lower value.

Line 33 with the content of extract? Does this content correspond with the volume of extract the volunteers consumed?

Line 35 talk about human metabolism is much better than human body.

Line 35 The sentence is confusing, please rewrite. 

Line 50 “ginseng product preparation” is not good at all, I recommend write sample preparation of ginseng compounds isolation or something like that.

Line 51 steaming is a process included inside of total process of extraction, please it is convenient to be more specific.

Line 66 The sentence is confusing

Figure 1. Please, unify the way of the radicals appear. What is the meaning of R1(C3), R3(C20)? Please explain that in the food note.

Line 89 it would be better to add another section of sample preparation separately.

Line 90 the correct form is positive ionization mode, please revise and change in all the paper.

Line 90 – 94 The paragraph is confusing how did you select the ions? Did you base on the literature? As you know, it is advisable to fragment the standards by infusion before to develop the method. Remember the fragmentation depends on many factors between them of the equipment.

Line 95 What the meaning LLE method? Please, you have to write complete the expression before using the acronyms the first time. 

Line 96 The sentence is confusing.

Line 100 “human blank plasma” Please, it is necessary to explain what is this.

Line 101 protein precipitation instead of precipitation

Line 103 What is the meaning of MTBE?

Line 107 What is the meaning of endogenous peaks?

Line 109 It is necessary a reference

Table 1 What is the meaning of Rc/Rb2? And What is the meaning of fragmentor? If it is fragmentor voltage you have to explain that in a foot table note with units, to make the read easier. When you write Protein precipitation or LLE, is it the only procedure using to sample preparation? Please explain that in the foot table note.

I think it would be better to add a section called Validation or Validation of the LC-MS method and in there grouping sections from 2.1.2 to 5. Indeed, remove 2.1 section.

Line 140 to 142 Is there any official guide you use as a reference? For example, the AOAC guide, if you use that kind of guide as a reference you reinforce the study. 

Line 156 Which are the four gisenosides? Please, write it.  I don’t understand.

Line 158 methanol precipitation?

Line 156 mass suppression values of 77.3 and 100.1 % are so great, you have to explain in depth. In my opinion, the small values of CV do not justify such high values in matrix effect.

Line 182 It will be more convenient to give the values in different way in comparison with the Table 6 if you are going to give a range, please you have to say “the values are ranged between … “or something similar.

Line 216 That conclusion deserves a reference

Line 245 The sentence is confusing, what method do you are talking about?

Line 246 to 251 this paragraph should not be here, please remove or add to the introduction.

Line 257 to 262 The aim of the study have to be at the end of the introduction. In the discussion section, you have to explain your results.

Line 285 it would be interesting you give a reason.

Line 359 Please explain in depth what is blank plasma samples.

Have the authors any reference in this section? Is there any guide to show a validated method to measure the matrix effect?

Author Response

Q1. Abstract: have 255 words more than 200 as the instructions of author indicates.

Answer> As the reviewer suggested, we reduced the abstract and the word count of Abstract is now 200.

Q2. Line 29 I suggest write the LOQ limits as a range of the highest to lower value.

Answer> The standard range of 13 ginsenoside [Rb1, Rb2, Rc, Rd, Re, Rf, Rg1, Rg3, Rh2, F1, compound K (CK), protopanaxadiol (PPD), and protopanaxatriol (PPT)] was all 0.5-200 ng/mL in human plasma and, therefore, the LLOQ of all the ginsenosides was 0.5 ng/mL. Therefore, we change the expression as follows:

[Lines 26-28] The calibration curve range for 13 ginsenosides was 0.5 - 200 ng/mL and the lower limit of quantitation was 0.5 ng/mL for all ginsenosides.

Q3. Line 33 with the content of extract? Does this content correspond with the volume of extract the volunteers consumed?

Answer> It means that the plasma concentrations of Rb1, Rb2, Rc, Rd, and Rg3 were correlated with the contents of ginsenosides Rb1, Rb2, Rc, Rd, and Rg3 in red ginseng extract.

Q4. Line 35 talk about human metabolism is much better than human body.

Answer> As the reviewer suggested, we changed “the formation of these ginsenosides in human body” into “the formation of these ginsenosides through the human metabolism”.

Q5. Line 35 The sentence is confusing, please rewrite.

Answer> We deleted the sentence to avoid the confusion of readers and because of the limitation of word count in Abstract.

Q6. Line 50 “ginseng product preparation” is not good at all, I recommend write sample preparation of ginseng compounds isolation or something like that.

Answer> As the reviewer suggested, we changed “method of ginseng product preparation such as steaming and extraction” into “preparation method of ginseng product such as steaming times and temperature and extraction method” [lines 45-46].

Q7. Line 51 steaming is a process included inside of total process of extraction, please it is convenient to be more specific.

Answer> As the reviewer suggested, we added the description regarding the preparation method and the changes in ginsenoside composition as follows during the revision:

[Lines 44-49] The content and types of ginsenosides vary depending on the preparation method of ginseng product such as steaming times and temperature and extraction method [9,10]. For example, ginsenosides Rg1 and Re are decreased but ginsenosides Rb1, Rb2, Rc, Rd, and Rg3 are increased after several hours of steaming and extraction. As results, ratio of PPD-type to PPT-type ginsenoside of Korean red ginseng extract was higher than that of Korean ginseng [10].

Q8. Line 66 The sentence is confusing

Answer> As the reviewer suggested, we revised as follows:

[Lines 75-83] In another study, ginsenoside PPD was analyzed after a single oral administration of a PPD 25 mg capsule with a calibration curve range of 0.1 – 100 ng/mL [16]. Ginsenoside Re was analyzed after a single oral administration of a Re 200 mg tablet with a calibration curve range of 0.5 – 200 ng/mL. The metabolite peaks of Rg1, Rg2, F1, Rh1, and PPT in human plasma and urine were also monitored following oral administration of Re tablet (200 mg) without quantification [12]. Our group simultaneously determined the plasma concentrations of the ginsenosides Rb1, Rb2, Rc, Rd, and CK in human subjects following single and 2-week repeated administration of 3 pouches of red ginseng product with a calibration curve range of 0.5 – 200 ng/mL [17].

Q9. Figure 1. Please, unify the way of the radicals appear. What is the meaning of R1(C3), R3(C20)? Please explain that in the food note.

Answer> We added carbon numbering to the core triterpene saponin structure in Figure 1 to clarify the meaning of R1(C3), R3(C20). 

Q10. Line 89 it would be better to add another section of sample preparation separately.

Answer> According to the reviewer’s suggestion, we divided into the MS/MS analysis and Sample preparation section during the revision [lines 103 and 112]

 Q11. Line 90 the correct form is positive ionization mode, please revise and change in all the paper.

Answer> According to the reviewer’s suggestion, we corrected the “positive mode” into “positive ionization mode” in the revised manuscript [line 104].

Q12. Line 90 – 94 The paragraph is confusing how did you select the ions? Did you base on the literature? As you know, it is advisable to fragment the standards by infusion before to develop the method. Remember the fragmentation depends on many factors between them of the equipment.

Answer> As we described in the Material section, we purchased 15 ginsenosides (Rb1, Rb2, Rc, Rd, Re, Rf, Rg1, Rg3, Rh1, Rh2, F1, F2, CK, PPD, and PPT) from the Ambo Institute (Daejeon, Republic of Korea) as authentic standard materials. The selected precursor and product ions of analytes and respective mass spectrometric conditions of 15 ginsenosides were optimized in our MS/MS system. Therefore, the values shown in Table 1 were the optimized conditions in our MS/MS system and m/z values of precursor and product ion from standard ginsenosides were also consistent with the references [19] and [20]. The expression in the manuscript was revised as follows:

[Lines 105-111] Table 1 shows the selected precursor and product ions of analytes and respective mass spectrometric conditions in the MS/MS stage of the ginsenosides, which were optimized based on the fragmentation patterns of precursor and product ions of target ginsenoside, the specificity of target ginsenoside compared to the other ginsenosides, and the consistency with the previously published findings [19,20]. Since ginsenoside Rb2 and Rc resulted in the same m/z values of precursor and product ion, these ginsenosides should be separated each other during the elution. Retention times were 5.7 min for Rb2 and 4.8 min for Rc (Table 1 and Figure 2B).

Q13. Line 95 What the meaning LLE method? Please, you have to write complete the expression before using the acronyms the first time.

Answer> Abbreviation “LLE” was provided with a full name at its first appearance “liquid-liquid extraction (LLE)” during the revision (Line 113).

Q14. Line 96 The sentence is confusing.

Answer> According to the reviewer’s suggestion, we revised the confusing sentence as follows:

[Lines 113-118] For sample preparation, both protein precipitation and liquid-liquid extraction (LLE) methods should be applied depend on the number of glycosylation of ginsenosides. For example, we used the protein precipitation method for ginsenosides glycosylated with more than two glucose units (i.e., Rb1, Rb2, Rc, Rd, Re, Rf, Rg1, Rg3, and F2; hydrophilic ginsenosides) and the LLE method for monoglycosylated ginsenosides and their aglycones (i.e., Rh1, Rh2, CK, PPD, and PPT; lipophilic ginsenosides)   

Q15. Line 100 “human blank plasma” Please, it is necessary to explain what is this.

Answer> Human blank plasma mean the human plasma withdrawn from human subjects who did not take ginseng or ginsenosides and it should be used for the validation of analytical method. To make the confusing sentence clear, we revised the sentence as follows:

[Lines 118-120] … based on the extraction recovery after sample preparation and the interference of endogenous peaks in human blank plasma (the plasma withdrawn from human subjects who did not take ginseng or ginsenosides).

Q16. Line 101 protein precipitation instead of precipitation

Answer> “precipitation” was revised as “protein precipitation” [line 121].

Q17. Line 103 What is the meaning of MTBE?

Answer> “MTBE” was corrected as “methyl tert-butyl ether (MTBE)” [line 123].

Q18. Line 107 What is the meaning of endogenous peaks?

Answer> Endogenous peaks refers as the unknown peaks in the human blank plasma detected at the same m/z as target ginsenosides analyzed in this study. In the specificity from method validation, we confirmed that these endogenous peaks did not interfere with the analysis of ginsenosides. We added the description of endogenous peak in the revised manuscript as follows:

[Lines 127-128] the endogenous peaks that detected at the same m/z as F2 and Rh1 in human blank plasma

Q19. Line 109 It is necessary a reference

Answer> According to the reviewer’s suggestion, we added the reference and the sentence was revised as follows:

[Lines 126-129] The ginsenosides F2 and Rh1 were excluded in the validation process because their peaks could not be completely separated from the endogenous peaks that detected at the same m/z as F2 and Rh1 in human blank plasma, and the peak response of F2 and Rh1 at LLOQ was less than five times the response of a blank sample [21].   

Q20. Table 1 What is the meaning of Rc/Rb2? And What is the meaning of fragmentor? If it is fragmentor voltage you have to explain that in a foot table note with units, to make the read easier. When you write Protein precipitation or LLE, is it the only procedure using to sample preparation? Please explain that in the foot table note.

Answer> Rb2/Rc mean ginsenoside Rb1 and Rc. To avoid confusion, we separately provide the MS/MS parameters of Rb2 and Rc in Table 1.

Fragmentor voltage (V) is the voltage difference between capillary and skimmer, which as defined in a foot note. Protein precipitation and LLE are the sample preparation methods that we used for the analysis of ginsenosides. Therefore, we added the unit of fragmentor voltage and collision energy and sample preparation method in the head of Table 1.  

Q21. I think it would be better to add a section called Validation or Validation of the LC-MS method and in there grouping sections from 2.1.2 to 5. Indeed, remove 2.1 section.

Answer> According to the reviewer’s suggestion, we changed the section title as follows:

2.1. MS/MS analysis

2.2. Sample praperations

2.3. Analytical method validation

2.3.1. Specificity and other subsections.

Q22. Line 140 to 142 Is there any official guide you use as a reference? For example, the AOAC guide, if you use that kind of guide as a reference you reinforce the study.

Answer> The analytical method was validated according to the FDA Guidance for Industry: Bioanalytical Method Validation (May 2018) and we added this as follows:

[Lines 134-136] The method was fully validated according to the FDA Guidance for Industry: Bioanalytical Method Validation (May 2018) [22] for its specificity, accuracy, precision, matrix effect and extraction recovery, and stability.

 Q23. Line 156 Which are the four gisenosides? Please, write it. I don’t understand.

Answer> 4 ginsenosides were specified as “the ginsenosides Rh2, CK, PPD, and PPT” in the revised manuscript [Line 174].

Q24. Line 158 methanol precipitation?

Answer> “methanol precipitation” was corrected as “protein precipitation” [line 178].

Q25. Line 156 mass suppression values of 77.3 and 100.1 % are so great, you have to explain in depth. In my opinion, the small values of CV do not justify such high values in matrix effect.

Answer> In our analytical method, ginsenosides Rh2, CK, PPD, and PPT were processed using LLE method that could remove endogenous compound such as lipids, phospholipids, and fatty acids, that may affect the ESI droplet desolvation process, therefore matrix effect would be high. However, protein precipitation does not completely remove the endogenous substances, such as lipids, phospholipids, and fatty acids, therefore ion suppression is quite common in ESI, whenever complex mixtures are studied. We tried to separate 9 ginsenosides (i.e., Rb1, Rb2, Rc, Rd, Re, Rf, Rg1, Rg3, and F2) that were processed using protein precipitation method with different retention time via the gradient elution. However, Re, Rf, and Rg1 that showed short retention time were highly affected by ion suppression. This could be attributed to the short retention time and sample preparation method (higher chance to ion suppression by endogenous lipids, phospholipids, and fatty acids). EMA guideline suggested that the analytical methods are valid when the CV of IS-normalised matrix effect from the 6 batch of matrix was less than 15%. In our case, CV of IS-normalised matrix effect for Re, Rf, and Rg1 was less than 15% and the matrix effect of Re, Rf, and Rg1 was similar over the 3 different QC levels. Therefore we concluded our analytical method was valid even though Re, Rf, and Rg1 had significant ion suppression according to the EMA guideline (Guideline on bioanalytical method validation. London: 2011).

We added the decision based on the EMA guideline in the revised manuscript:

[Lines 176-183] The matrix effect of ginsenosides Re, Rf, and Rg1 was in the range of 7.0% – 19.5%, suggesting that Re, Rf, and Rg1 showed significant signal suppression during the ionization and protein precipitation process; however, the values of CV of Re, Rf, and Rg1 was less than 15% and the matrix effect of Re, Rf, and Rg1 was similar for the three different QC levels with an acceptable CV, and 10 other ginsenosides showed no significant interference during ionization and sample preparation. According to the EMA guideline [23], we concluded our analytical method was acceptable even though Re, Rf, and Rg1 had significant ion suppression.

Q26. Line 182 It will be more convenient to give the values in different way in comparison with the Table 6 if you are going to give a range, please you have to say “the values are ranged between … “or something similar.

Answer> As the reviewer suggested, we change the expression of the content of ginsenoside as a range.

[Lines 203-207] The most abundant ginsenoside was Rb1 (18.8-23.6 mg/day), followed by Rb2, Rc, Rd, and Rg3 (12.9–5.9 mg/day). The abundance of Re, Rh1, and Rg1 was 1.6–6.6 mg/day. The daily intake of PPT-type ginsenosides was lower than that of PPD-type ginsenosides. The values of daily intake of PPD-type ginsenosides are ranged between 50.2-64.7 mg/day and those of PPT-type ginsenoside are ranged between 11.2-14.9 mg/day.

Q27. Line 216 That conclusion deserves a reference

Answer> As the reviewer suggested, we added reference and added some description on this reference as follows:

[Lines 237-240] Lin et al. reported that 40 metabolites of PPD was identified in human plasma and urine and the major metabolites of PPD were hydroxylated form in human body through phase I hepatic metabolism [19].

Q28. Line 245 The sentence is confusing, what method do you are talking about?

As the reviewer suggested, we revised the corresponding paragraph as follows to avoid confusion of the description and moved this paragraph to Introduction to remove redundancy.

[Lines 61-71] In human study, there are some reports on the analysis of ginsenosides but they used large volume of plasma or provided limited concentrations on ginsenoside because of the high lower limit of quantitation (LLOQ). For example, the plasma concentration of Rb1 and CK following single oral administration of 10 g of American ginseng powder was investigated. In this study, Rb1 and CK in 0.7 mL of plasma samples were extracted using a solid-phase extraction procedure and detected by time-of-flight mass spectrometry coupled with ultra-high pressure liquid chromatography [13]. Another study developed a simultaneous analysis method for Rb1 and Rg1 in human plasma by LC-MS/MS. In this study, for the analysis of Rb1 and Rg1, 100 μL of human plasma was subjected to protein precipitation and analyzed with a calibration curve range of 10 – 1000 ng/mL [14]. The ginsenoside Rb1 was detected but Rg1 was not detected. This could be attributed to the low plasma concentration of Rg1 after oral administration of the ginseng product (1.5 g/day) [14].

Q29. Line 246 to 251 this paragraph should not be here, please remove or add to the introduction.

Answer> According to the reviewer’s suggestion, this analytical part moved into introduction [lines 72-89].

Q30. Line 257 to 262 The aim of the study have to be at the end of the introduction. In the discussion section, you have to explain your results.

Answer> We deleted the sentence regarding the purpose of this study.

Q31. Line 285 it would be interesting you give a reason.

Answer> Thank you for the reviewer’s valuable comments. However, at present, the time-dependent gut metabolism of ginsenosides in human intestine has never been investigated, therefore we speculated time-dependent gut metabolism of ginsenoside from the plasma concentration and Tmax of ginsenosides and their deglycosylated metabolites.

We added the results (including Figure 4) and discussion regarding the time dependent metabolism and absorption as follows:

[Lines 241-252] To explain time-dependent metabolism and absorption of ginsenosides, the plasma concentrations of ginsenosides at absorption phase (from 4 to 10 h) depend on the deglycosylation states was shown in Figure 4. The plasma concentrations of Rb1, Rb2, Rc, and Rd, tri- and tetraglycosylated ginsenosides, were stable for 4 - 10 h of post dose (Figure 4A), suggesting the stable absorption and slow elimination process. The plasma concentrations of Rg3 was decreased along with increasing time (4 – 10 h) but the monoglycosylated ginsenosides Rh2 and CK, metabolites from Rg3 and F2, increased over time (Figure 4B and 4C), suggesting the gut metabolism from Rg3 to Rh2 during the absorption stage. The delayed absorption of Rh2, CK, and PPD indicated that formation and absorption of Rh2, CK, and PPD might occur in the lower part of intestine. On the other hand, the formation and absorption of PPT was faster than PPD (Figure 4D), suggesting the rapid metabolism of PPT-type ginsenosides in human intestine and it partly attributed to the absence of Re and Rg1 in human plasma despite of the higher content in Korean red ginseng extract.

[Lines 301-308] It is possible that Re and Rg1 are metabolized to PPT by intestinal microbiota before the absorption occur [11,23,24] and biotransformation of PPT could be faster than the formation rate of PPD. However, we should note that the time-dependent gut metabolism of ginsenosides in human intestine has never been investigated, therefore we speculated time-dependent gut metabolism of ginsenoside from the plasma concentration and Tmax of ginsenosides and their deglycosylated metabolites. Particularly, for CK concentration, large inter-subject variation was shown in Figure 4B and previous publication [17]. This variability could be attributed to inter-subject variable metabolism related to the intestinal microbiota [25] and further studies should focus on the characterization of microorganisms that produce it and the potential beneficial effects of this metabolite.

Q32. Line 359 Please explain in depth what is blank plasma samples.

Answer> As we explained previously in Q15, human blank plasma mean the human plasma withdrawn from human subjects who did not take ginseng or ginsenosides. Since we added explanation on human blank plasma in the Results section [Line 118-120], we leave it as it is. During the revision, we realized that we confusingly used human blank plasma and blank human plasma by mistake. We unified as human blank plasma throughout the whole manuscript. We ask generous understanding on this issue.

Q33. Have the authors any reference in this section? Is there any guide to show a validated method to measure the matrix effect?

Answer> Matrix effect was measured according to the following reference “Matuszewski BK, Constanzer ML, et al. Strategies for the assessment of matrix effect in quantitative bioanalytical methods based on HPLC-MS/MS. Analytical Chemistry. 2003; 75(13):3019–3030”. We added this reference to the revised manuscript [Lines 380-382].

Reviewer 4 Report

The manuscript by Jin et al. describes a rapid and reproducible analytical method to analyze ginsenosides in human plasma. The manuscript is very well written and results are nicely presented for readers to interpret the data. The authors start by carrying out systematic characterization of the main constituents and present two different sample preparation method (protein precipitation and LLE) for comprehensive analysis of the 13 ginesonsides. The pharmacokinetics study in plasma from a human cohort provides significant results for scientific community involved in clinical research.

Specific Comments:

(a)    Page 1, Line 67-68 – Add reference.

(b)    Page 8, Section 2.1.6. It is not uncommon in a clinical setting for a blood specimen to sit outside for several hours before it is being processed to plasma and the authors did not carry out any evaluation of this factor. While they did carry out stability of spiked-in ginsenosides in plasma, it will be good if the authors can comment on the impact of blood processing time to stability of these compounds.  For example, if the blood samples processed in pharmacokinetic study were left at room temperature for several hours before processing into plasma component, will the results be any different from those reported in the manuscript.

(c)    Figure 2 – Provide a high resolution image of the MRM chromatograms.

Author Response

Q1. Page 1, Line 67-68 – Add reference.

Answer> As the reviewer suggested, we added reference during the revision [lines 77-80].

Q2. Page 8, Section 2.1.6. It is not uncommon in a clinical setting for a blood specimen to sit outside for several hours before it is being processed to plasma and the authors did not carry out any evaluation of this factor. While they did carry out stability of spiked-in ginsenosides in plasma, it will be good if the authors can comment on the impact of blood processing time to stability of these compounds.  For example, if the blood samples processed in pharmacokinetic study were left at room temperature for several hours before processing into plasma component, will the results be any different from those reported in the manuscript.

Answer> As the reviewer pointed out, the blood samples were collected using heparinized vacutainer and, upon receipt of the blood sample, they were centrifuged (3000 x g, 4 oC, 10 min). The plasma samples were stored at -80 oC freezer until analysis. Therefore, stability issues focus on the stability of analytes during the storage in the freezer, sample preparation process, and analysis time after the samples were processed. Our analytical method was validated according to the FDA Guidance for Industry: Bioanalytical Method Validation (May 2018) for its specificity, accuracy, precision, matrix effect and extraction recovery, and stability. According to the guideline, stability item includes short-term stability (Bench-top stability), post-preparative stability (Autosampler stability), and freeze-thaw stability. We used three quality control samples of high, middle, and low concentrations of analytes and evaluated short-term stability (Bench-top stability), post-preparative stability (Autosampler stability), and freeze-thaw stability. Stability results fell within the acceptance criteria and thus we could conclude that the stability of ginsenoside could be guaranteed during the analysis.

According to the reviewer’s suggestion, we revised the result section as follows:

[Lines 193-195] Therefore, the 13 ginsenosides in human plasma samples had no stability issues during the storage in the freezer, sample preparation process, and analysis time after the samples were processed, as demonstrated by the three stability tests.

Q3. Figure 2 – Provide a high resolution image of the MRM chromatograms.

Answer> As the reviewer suggested, we increased the resolution of Figure 2 during the revision.
